Variable level of genetic dominance controls important agronomic traits in rice populations under water deficit condition

Hassan Hamada M. 1
Hadifa Adel A. 1
El-leithy Sara A. 1
Batool Maria 2
http://orcid.org/0000-0003-2087-6555 Sherif Ahmed 1 2
http://orcid.org/0000-0001-5022-1555 Al-Ashkar Ibrahim 3
Ueda Akihiro 4
Rahman Md Atikur 5
Hossain Mohammad Anwar 6 anwargpb@bau.edu.bd
http://orcid.org/0000-0003-3477-762X Elsabagh Ayman 7 aymanelsabagh@gmail.com
1 Department of Rice Research, Field Crops Research Institute, Agricultural Research Center (ARC) , Giza , Egypt
2 College of Plant Science & Technology, Huazhong Agricultural University , Wuhan , China
3 Department of Plant Production, College of Food and Agriculture, King Saud University , Riyadh , Saudi Arabia
4 Graduate School of Integrated Sciences for Life, Hiroshima University of Economics , Hiroshima , Japan
5 Grassland and Forage Division, National Institute of Animal Science, Rural Development Administration , Cheonan , Republic of Korea
6 Department of Genetics and Plant Breeding, Bangladesh Agricultural University , Mymensingh , Bangladesh
7 Department of Agronomy, Faculty of Agriculture, Kafrelsheikh University , Kafr El-Shaikh , Egypt
Abd El-Moneim Diaa
Electronic publication date: 2023 Feb 13
Publication date: 2023
Volume: 11
Electronic Location ID: e14833
Received 2022 Sep 19; Accepted 2023 Jan 10
Copyright: © 2023 Hassan et al.
Copyright year: 2023
Copyright holder: Hassan et al.
License: This is an open access article distributed under the terms of the Creative Commons Attribution License, which permits unrestricted use, distribution, reproduction and adaptation in any medium and for any purpose provided that it is properly attributed. For attribution, the original author(s), title, publication source (PeerJ) and either DOI or URL of the article must be cited.
License URL: https://creativecommons.org/licenses/by/4.0/

Keywords: Oryza sativa L., Additive, Dominance, Heritability, Genetic advance, Water deficiency, Root, Grain yield

Funding: King Saud University, Riyadh, Saudi Arabia (RSP2023R298) This work was funded by the Researchers Supporting Project number (RSP2023R298), King Saud University, Riyadh, Saudi Arabia. The funders had no role in study design, data collection and analysis, decision to publish, or preparation of the manuscript.

==============================
Plant hybridization is an important breeding technique essential for producing a genotype (hybrid) with favorable traits (e.g., stress tolerance, pest resistance, high yield potential etc.) to increase agronomic, economic and commercial values. Studying of genetic dominance among the population helps to determine gene action, heritability and candidate gene selection for plant breeding program. Therefore, this investigation was aimed to evaluate gene action, heritability, genetic advance and heterosis of rice root, agronomic, and yield component traits under water deficit conditions. In this study, crossing was performed among the four different water-deficit tolerant rice genotypes to produce better hybrid (F1), segregating (F2) and back-cross (BC1 and BC2) populations. The Giza 178, WAB56-204, and Sakha104 × WAB56-104 populations showed the better physiological and agronomical performances, which provided better adaptability of the populations to water deficit condition. Additionally, the estimation of heterosis and heterobeltiosis of some quantitative traits in rice populations were also studied. The inheritance of all studied traits was influenced by additive gene actions. Dominance gene actions played a major role in controlling the genetic variance among studied traits in both crossed populations under well-watered and drought conditions. The additive × additive type of gene interactions was essential for the inheritance of root length, root/shoot ratio, 1,000-grain weight, and sterility % of two crossed populations under both conditions. On the contrary, the additive × dominance type of gene interactions was effective in the inheritance of all studied traits, except duration in Giza178 × Sakha106, and plant height in Sakha104 × WAB56-104 under water deficit condition. In both crosses, the dominance × dominance type of gene interactions was effective in the inheritance of root volume, root/shoot ratio, number of panicles/plant and 1,000-grain weight under both conditions. Moreover, dominance × dominance type of gene interaction played a major role in the inheritance of root length, number of roots/plant, plant height, panicle length, number of filled grain/panicle and grain yield/plant in Giza178 × Sakha106 under both conditions. The studied traits in both crossed populations indicated better genetic advance as they showed advanced qualitative and quantitative characters in rice populations under water deficit condition. Overall, our findings open a new avenue of future phenotypic and genotypic association studies in rice. These insights might be useful to the plant breeders and farmers for developing water deficit tolerant rice cultivars.

Introduction

Rice (Oryza sativa L.) is an important cereal food crop, and one of the vital sources of human nutrition, and global food security (Kondhia, Tabien & Ibrahim, 2015). Rice is considered as major source of food stuff in Asia and Africa, and it constitutes cover approximately 50–80% of daily calories (Futakuchi, Manful & Sakurai, 2013; Seck et al., 2012). Rice grain provides approximately 23% of energy as well as 16% of protein per capita globally (Ye et al., 2000). Rice is cultivating globally more than 160 million hectares areas are using in rice production that annually producing about 740 million tons rice (Kumar et al., 2019). In Asia, 79 million hectares of irrigated land have been used for rice cultivation annually, producing nearly 75% of the rice in the world. Thus, current and future global food security will be highly dependent on the irrigated rice production system (Fahad et al., 2019). The demand for rice consumption increases annually due to the increasing world population; however, rice production has been facing extreme climatic challenges causing yield instability. Satisfying the gap between rice supply and demand by uplifting production is one of the key strategies to endure rice production with declining land and water reserves and ensure the stability of global food security for the expected population growth of 9.8 billion by 2050 (United Nations, 2019).Therefore, global rice production needs to increase with increased tolerance in plants due to global food demand and adaptation in plants to changing climate conditions.

During irrigation, high levels of soil deepwater requires for growth and development (Carrijo et al., 2019). Unfortunately, rice faces water shortage due to its short fibrous root system, while deep root containing cultivar is more tolerance to drought (Kim et al., 2020). Water scarcity or drought is one of the major environmental factors hindering rice production (Melandri et al., 2020). Climate change significantly alters the efficiency of plant adaptation to changing environments (Sabagh et al., 2020; Raza et al., 2022b). Multiple abiotic stresses significantly decline plant growth, physiological efficiency, and agricultural productivity (Rahman et al., 2021; Khan et al., 2022). Water deficit is one of the major limitations that dramatically induces water scarcity in short root system containing cereals (Kim et al., 2020). The damage caused by water deficit to rice growth and development is multifaceted, and the mechanism of rice responses is very complex. Therefore, improvement of abiotic stress tolerance in plants through a breeding program is highly desirable (Lee et al., 2017). Screening of stress tolerance in plant cultivar is also considered as important traits (Rahman et al., 2015, 2022). Plant can cope with water deficit by different strategies, including stress mitigation, avoidance, and tolerance. Thus upland rice can escape water deficit by early maturity, avoid it through drought-induced elongation of roots to reach comparatively deeper moisture zones, or tolerate it through reducing transpiration losses by leaf rolling, early closure of stomatal openings and cuticular resistance (Virmani & Ilyas-Ahmed, 2007). Furthermore, plant enhances stress tolerance by scavenging reactive oxygen species (ROS) through boosting of metabolites, enzymatic and non-enzymatic antioxidant system (Raza et al., 2022a; Kabir et al., 2021; Sabagh et al., 2021). The grain yield and its components traits, as well as root traits have been affected by water-limited conditions, as observed in the sensitive varieties. While, the tolerant varieties that have some mechanisms to resist drought stress were slightly affected by water deficit (Hassan, El-Abd & El-Baghdady, 2011; Hassan, El-Khoby & El-Hissewy, 2013). Recently, root system architecture of cereals has been targeting for modern breeding program to develop drought tolerant varieties (McGrail & McNear, 2021). However, root system traits are important in maintaining plant productivity under drought stress. One is the overall root system size that is related to the acquisition of water and nutrients from soil and should be accompanied with a balanced leaf surface area ratio (Diaz-Espejo et al., 2012). The architecture of the root system that are important to acquire drought tolerance include root diameter, root tissue density and other parameters such as specific root length (SRL) or specific surface area (SSA) that are measures of root length and surface area per dry mass, respectively (Comas et al., 2013). Another important feature in roots of drought-tolerant plants is the roots fitness, potential of water and nutrient uptake (Peterson, Murrmann & Steudle, 1993), and sufficient density and length of root hairs that enlarge the surface of contact between soil and roots (Wasson et al., 2012). Additionally, the development of a root system under drought conditions is controlled by a wide range of molecular mechanisms, including signalling of the soil water status toward the shoot and changes in the regulation of gene expression, and subsequent control of molecular pathways. Therefore, understanding this molecular network of roots and its regulatory mechanisms is important for enhancing water deficit tolerance in plants (Janiak, Kwaśniewski & Szarejko, 2015).

Hybridization is a useful approach for enhancing stress tolerance in plants and ensuring global food security in changing environments. Screening of genetic dominance in hybrids or plant population is important to determine the gene effects, heritability and selection of novel genes that help to develop stress tolerance in plants through plant breeding program. Furthermore, various water-saving techniques have been used in rice cultivation, including alternate wetting and drying (AWD), continuous soil saturation, irrigation at constant soil moisture tensions ranging from 0 to 40 kPa and irrigation every 1 to 5 days after the disappearance of standing water. These types of water management systems are known as partial aerated rice systems (PARSs); according to the results of these studies, water-saving technologies can increase crop water productivity without significantly reducing crop yield (Prasad, 2011). The effectiveness of selection between the offsprings with different genetic values is the primary determinant of genetic improvement. Understanding of generic actions, heterosis, heritability, and genetic components for crops provides the information required to select the best breeding approaches for variety development programs (Ammar et al., 2014). Genetic actions are the additive and dominant influences, as well as their interactions are linked to the value of breeding (Xu, 2022). Additionally, genetic analysis using generation mean analysis (GMA) has been used to estimate genetic actions that control quantitative traits and breeders will benefit from understanding additive, controlling and cognitive influences when designing the most appropriate breeding methods for developing a new cultivar (Sher et al., 2012). The GMA consists of six groups/generations for estimation of gene actions and associations: parent 1 (P1), parent 2 (P2), first offspring (F1), second offspring (F2), first offspring from posterior crossover with recurrent parent 1 (BC1), and the first child of backcross with recurrent parent 2 (BC2). Rice has been used for root properties that are controlled by additive and control genetic actions in selected aromatic rice crosses (Kumari, Kumar & Kumar, 2019). The influence of additive and dominant genes was reported to be significant for other morphological and agronomic traits such as duration, plant height, panicle length, number of panicles/plant, number of filled grains/panicle 1,000-grain weight, sterility % and grain yield/plant. In rice water deficit, GMA has not yet been used to estimate the genetic effect of quantitative traits (Kumari, Kumar & Kumar, 2019).

Plant organs, such as leaves and roots, coordinate defense mechanisms (internal or external) in response to abiotic stress (Nadarajah & Kumar, 2019). The root is the first sensing organ to experience water deficit stress because the water level declines due to insufficient water availability in the soil (Koevoets et al., 2016). Also, physiological and agronomic traits are affected by genetic and various environmental factors (Zhang et al., 2018). However, this study was aimed to evaluate whether variable level of genetic dominance controls important agronomic traits in rice populations under water deficit condition. We have also used breeding and selection approaches to screen better water deficit tolerant rice population.

Materials and Methods

Crossing and development of F1 and F2 populations

The current study was carried out at the Sakha Agricultural Research Station (31°05′17″N, 30°56′44″E), Egypt during three consecutive seasons: 2019, 2020, and 2021. Two rice populations Giza178 × Sakha106 (P1 × P2) and Sakha104 × WAB56-104 (P3 × P4) were crossed to produce F1 hybrids in 2019 according to the method (Abd El-Hadi et al., 2012). The varieties P1: Giza178 and P4: WAB56-104 were tolerant to water deficit, while P2: Sakha106 and P3: Sakha104 were sensitive (Hassan, El-Abd & El-Baghdady, 2011; Hassan, El-Khoby & El-Hissewy, 2013) (Table 1 and Table S1). In the 2020 season, parents and F1 hybrid seeds were planted in experimental field. The F1 generation was back crossed with its respective parents to produce the first backcross (F1 × P1) and second backcross (F1 × P2) generations. Simultaneously, F1 hybrid seeds were produced through a cross between the two parents. Subsequently, selfing the F1 plants were performed and produced F2 seeds. Seeds from the six populations including first parent (P1), second parent (P2), first-generation (F1), second-generation (F2), first backcross (BC1) and second backcross (BC2) were sown in the dry seedbed during the 2021 season (Fig. 1). Seedlings were transplanted after 30 days of sowing. Therefore, the total experiments were conducted following a randomized complete block design (RCBD) with four replications, in which each replicate consisted of seven rows. All agricultural practices, including sowing (the plants were sown as single plant in a row, and each row consisted 25 plants (1 m2) with 20 × 20 cm planting spaces). Several combinations of fertilization (100 N/ha, nitrogen fertilizer was used as urea form 46.5% N in two splits; 2/3 part was applied as basal and mixed in dry soil before flooding irrigation, and 1/3 part was applied at panicle initiation stage), and weed control were properly maintained. Plants were watered at field capacity until the growth of 2 weeks. Drought stress treatment was given after 2 weeks of plant growth by withdrawing water. One more set of plants was maintained with regular watering to consider as control. Data were recorded at the maturity stage; 30 plants for each P1, P2 and F1, along with a total 120 plants from each BC1, BC2, and 200 plants from the F2 generation were considered for data analyses.

Table 1 Assessment of MP, BP and degree of dominance of root, agronomic and yield traits for the two studied crosses under both conditions.

Characters	Cr.	Heterosis %	Degree of dominance	
MP	BP			
N	D	N	D	N	D	
Root length (cm)	I	63.67**	46.71**	44.24**	39.17**	−8.62	−6.98	
II	37.27**	20.88**	28.52**	10.16**	−12.43	−20.18	
Root volume (cm3)	I	104.04**	173.74**	46.94**	94.10**	4.23	2.67	
II	78.05**	153.18**	48.33**	75.89**	−3.48	−3.89	
No. of roots/plant	I	59.75**	123.31**	34.77**	118.49**	55.92	3.22	
II	69.50**	33.42**	45.56**	29.45**	−10.88	−4.22	
Root/shoot ratio (%)	I	97.63**	576.06**	28.83**	10,17.74**	14.57	1.82	
II	54.35**	13.00**	20.40**	6.84**	2.26	−1.92	
Duration (day)	I	14.88**	−0.88	20.71**	3.31*	−0.22	−3.08	
II	0.64	2.45	3.87**	2.74	8.80	0.20	
Plant height (cm)	I	18.74**	4.56**	22.12**	6.00**	3.37	−6.76	
II	26.29**	−8.09**	27.42**	−4.26**	−2.02	40.23	
Panicle length (cm)	I	11.69**	16.31**	7.19**	7.55**	2.00	−2.78	
II	16.84**	4.30**	11.49**	3.75**	−8.11	−3.48	
No. of panicles/plant	I	28.94**	54.28**	8.83**	20.06**	1.90	1.56	
II	21.58**	44.38**	12.26**	18.80**	−2.06	−2.59	
No. of filled grains/panicle	I	−22.30**	−26.74**	−30.16**	−38.72**	−1.36	−1.98	
II	24.34**	8.48**	5.96**	1.33	−1.20	−1.40	
1,000-grain weight (g)	I	5.98**	14.72**	−10.16**	−1.48*	−0.96	−0.33	
II	−0.16	3.20**	−12.84**	−11.94**	0.18	−0.01	
Sterility %	I	632.28**	384.39**	1,140.40**	638.29**	−11.17	−15.43	
II	286.12**	111.60**	378.96**	137.05**	−15.24	−14.76	
Grain yield/plant (g)	I	16.20**	19.65**	6.46**	2.96**	1.21	1.77	
II	10.71**	23.70**	2.77**	10.23**	−1.93	−1.38	
Note:

Cross P1 × P2, Giza178 × Sakha106 and cross P3 × P4, Sakha104 × WAB56-104; N, normal and D, water deficit condition. One asterisk (*) and two asterisks (**) indicates significant at 5% and 1% level of probability.

Figure 1 Schematic map shows the hybridization, crossing and back crossing process for three years to get the basic generation.

P1, first parent; P2, second parent; F1, first-generation; F2, second-generation; BC1, first back-cross and BC2, second backcross.

Measurement of traits

Several agronomic traits of control and treated plants were measured after terminating drought stress. The root length of rice plants was measured using a centimeter (cm) scale. The number of roots per plant was calculated while plants were at the maximum tillering stage. The volume of the roots/plant was determined using cubic centimeters (cm3) by immersing the roots in a measuring cylinder tube filled with water. The proportion of the root dry weight and the shoot dry weight were measured at the maximum tillering stage using the following formula:

Root/shootratio=Rootdryweight(g)Shootdryweight(g)

However, the agronomic and yield-related traits, including plant height (cm), number of panicles/plant, panicle length (cm), 1,000-grain weight (g), number of filled grains/panicle, sterility % and grain yield/plant (g) were recorded according to the method of IRRI (2002).

Estimation of heterosis

Heterosis was calculated using the equation given previously (Anis et al., 2016), and appropriate LSD values were computed to test the significance of heterotic effects according to the method (Wynne, Emery & Rice, 1970). The relative of potency ratio (P) was calculated using the formula used previously (Mather & Jinks, 1971).

Determination gene actions

Scaling test for the model adequacy

The scaling tests (A, B and C) were calculated for each trait to determine the adequacy of the additive-dominance model or the presence of non-allelic gene interaction according to Evans, Gillespie & Martin (2002). Three tests (scales), A, B, and C were formed by maintaining zero within the limits of their standard errors. However, the significance of these scales was recorded to indicate the presence of non-allelic interactions. The significance from zero was tested using a t-test and the variance means for these estimates were obtained according to Evans, Gillespie & Martin (2002).

A=P1+F1−2BC1=1/2[(i)−(j)+(l)]

B=P2+F1−2BC2=1/2[(i)+(i)+(l)]

C=P1+P2+2F1−4F2=2(i)+(l)

where: VA, VB, and VC represent the variances of different effects and VP1, VP2, VF1, VF2, VBC1 and VBC2 represent the variances of different population means for the each cross.

Gene effects and allelic interactions

Estimates of various gene effects, allelic interaction, and their significance test were computed by a six parameter model (Hayman, 1958; Jinks & Jones, 1958).

m=Mean=F2

d=Additiveeffect=BC1−BC2

h=Dominanceeffect=2BC1+2BC2+F1−4F2−(1/2)P1−(1/2)P2

i=Additive×Additivegeneticinteraction

=2BC1+2BC2−4F2

j=Additive×Dominancegeneticinteraction

=2BC1−P1−2BC2+P2

l=Dominance×Dominancegeneticinteraction

=P1+P2+2F1+4F2−4BC1−4BC2

where: The parameters m, d, h, i, j, and l refer to mean effects, additive, dominance, additive × additive, additive × dominance, and dominance × dominance gene effects, respectively, whenever the phenotypic variance for each character was partitioned into additive (D), dominance (H) and environmental (E) variances using the method as described by Hayman (1958) as follows:

E=1/3(VP1+VP2+VF1)

D=4VF2−2(VBC1+VBC2)

H=4(VF2−1/2VD−VE)

Estimation of heritability and calculation of genetic advance upon selection (GS)

Broad sense heritability (h2b) and narrow-sense heritability (h2n) were calculated according to Powers (1950) and Warner (1952), respectively. Calculations of the genetic advance are expected (GS) and predicted (GS percent) values which were done following the method (Johnson, Robinson & Comstock, 1955).

Statistical analysis

The data related to all quantitative traits were subjected to standard statistical analysis. The analysis of variance (ANOVA) was carried out using Statistix software (version10).0. The significant difference of multiple comparison were analyzed using Duncan’s multiple range test (DMRT). The differences at P < 0.05, and P < 0.01 were considered as significant. GraphPad Prism software (version 9.0) was used to create the graphic presentation.

Results

Variations of traits in populations under normal and water deficit conditions

The phenotypic differences of root traits and tillering ability at the maximum tillering stage for Giza178 × Sakha106 (P1 × P2) and Sakha104 × WAB56-104 (P3 × P4) under normal and water deficit conditions were shown in Fig. 2. The results of the mean performance of the basic generation, i.e., P1, P2, F1, F2, BC1 and BC2, showed the wide differences of traits between the two parents under normal and water deficit conditions, respectively (Figs. 3A–3G, Tables S2 and S3). However, the mean values of root traits (Fig. 3), and other physiological and agronomic and yield traits were exhibited variations between the control and water deficit conditions (Figs. 4A–4G).

Figure 2 Phenotypic differences of root traits and tillering ability at the maximum tillering stage for two studied crosses under control and water deficit conditions.

Abbreviations, P1, female parent (P1, Giza178 and P3, Sakha104), P2, male parent (P2, Sakha106 and P4, WAB56-104) C1, Giza178 × Sakha106 (P1× P2) and C2, Sakha104 × WAB56-104 (P3 × P4). (A) C1 under normal condition, (B) C1 under water deficit condition, (C) C2 under normal condition, and (D) C2 under water deficit condition.

Figure 3 Means of six populations for rice root characters in two studied crosses under normal (N) and water deficit (D) conditions.

(A and B) root length (cm); (C and D) root volume (cm3); (E and F) number of roots/plant and (G and H) root/shoot ratio for cross (P1 × P2) and cross (P3 × P4), respectively. Bars represent the ± SE of four replicates. Letters on vertical bars represent significant differences between populations for the cross (P1 × P2) and cross (P3 × P4), respectively, according to Duncan’s multiple range test (DMRT) at P < 0.05. Asterisks indicate significant differences from normal and water deficit conditions, respectively (ns, not significant, *P < 0.05, **P < 0.01; Student’s t-test). Abbreviations, P1, female parent (P1, Giza178 and P3, Sakha104), P2, male parent (P2, Sakha106 and P4, WAB56-104) C1, Giza178 × Sakha106 (P1 × P2) and C2, Sakha104 × WAB56-104 (P3 × P4).

Figure 4 Means of six populations for rice vegetative characters in two studied crosses under normal (N) and water deficit (D) conditions.

(A and B) duration (day); (C and D) plant height (cm); (E and F) panicle length (cm) and (G and H) sterility % for cross (P1 × P2) and cross (P3 × P4), respectively. Bars represent the ± SE of four replicates. Letters on vertical bars represent significant differences between populations for the cross I and cross (P3 × P4), respectively, according to Duncan’s multiple range test (DMRT) at P < 0.05. Asterisks indicate significant differences from normal and water deficit conditions, respectively (ns, not significant, *P < 0.05, **P < 0.01; Student’s t-test). Abbreviations, P1, female parent (P1, Giza178 and P3, Sakha104), P2, male parent (P2, Sakha106 and P4, WAB56-104) C1, Giza178 × Sakha106 (P1 × P2) and C2, Sakha104 × WAB56-104 (P3 × P4).

Both in first (P1 × P2), and second cross (P3 × P4) populations, the root length was significantly higher in F1 than the rest of the populations (Figs. 3A and 3B). Root volume and roots/plant were also significantly higher in F1 of (P1 × P2) and (P3 × P4) crosses (Figs. 3B–3E). The root/shoot ration was significantly higher in F1 of (P1 × P2), while it was significantly declined in P4 of (P3 × P4) cross (Figs. 3G and 3H). No significant variation found in duration (day) in first (P1 × P2), and second cross (P3 × P4) (Figs. 4A and 4B). Plant high was declined by water deficit, the highest plant height was observed in F1 of (P1 × P2), compared to (P3 × P4) cross (Figs. 4C and 4D). However, no significant variation found in case of first (P1 × P2), and second cross (P3 × P4) populations except P2 of (P1 × P2) cross (Figs. 4E and 4F). Interesting, sterility % significantly influenced by water deficit (D) stress, the sterility % was highly significant in F1 of (P1 × P2) and (P3 × P4) crosses compared to other populations (Figs. 4G and 4H). The number of panicles/plant was in P1 and P2 of first (P1 × P2), and second cross (P3 × P4), respectively (Figs. 5A and 5B), while it was higher in F1 under D stress. The field grain/panicle was declined significantly in F2 of (P1 × P2) and (P3 × P4) crosses (Figs. 5C and 5D). The 1,000-grain weight was reduced under D stress compared to control. However, the 1,000-grain weight almost consistent in first (P1 × P2), and second cross (P3 × P4) populations under D stress but declined significantly in P1 of (P1 × P2), and P4 of (P1 × P2) cross, respectively (Figs. 5E and 5F). Additionally, the grain yield was higher in F1 of (P1× P2) and (P3 × P4) crosses (Figs. 5G and 5H).

Figure 5 Means of six populations for rice yield and its components characters in two studied crosses under normal (N) and water deficit (D) conditions.

(A and B) No. of panicles/plant; (C and D) no. of filled grains/panicle; (E and F) 1,000-grain weight (g) and (G and H) grain yield/plant (g) for cross (P1 × P2) and cross (P1 × P2), respectively. Bars represent the ± SE of four replicates. Letters on vertical bars represent significant differences between populations for the cross (P1 × P2) and cross (P1 × P2), respectively, according to Duncan’s multiple range test (DMRT) at P < 0.05. Asterisks indicate significant. Abbreviations, P1, female parent (P1, Giza178 and P3, Sakha104), P2, male parent (P2, Sakha106 and P4, WAB56-104) C1, Giza178 × Sakha106 (P1 × P2) and C2, Sakha104 × WAB56-104 (P3 × P4).

Heterosis and degree of dominance

Our results clearly showed that all root traits had significantly positive estimates of heterosis and heterobeltiosis (i.e., root length, root volume, number of roots/plant and root/shoot ratio) under both conditions in both crosses. Additionally, our investigation illustrated the significantly positive estimates of heterosis as a deviation from MP and BP for panicle length, number of panicles/plant, sterility % and grain yield/plant in both crosses under normal and water deficit conditions, while the plant height in (P1 × P2), number of filled grains/panicle in (P3 × P4) as a deviation from MP and BP and 1,000-grain weight in both crosses were highly significant positive estimates of heterosis as a deviation from MP (Table 1).

Moreover, plant height in (P3 × P4) under water deficit condition and number of filled grains/panicle in (P1 × P2) under both conditions were highly significant that negative estimates for heterosis as a deviation from MP and BP. In contrast, highly significant negative heterosis was estimated for 1,000-grain weight as a deviation from BP in both crosses. The other remaining traits showed non-significant positive and negative mean heterosis as a deviation from MP and BP in (P1 × P2) and (P3 × P4) under normal and water deficit conditions.

The degree of dominance was higher than unity (±1.0) for most of the studied traits, including root length and root volume, number of roots/plant, root/shoot ratio, plant height, panicle length, number of panicles/plant, number of filled grains/panicle, sterility % and grain yield/plant in both crosses under normal and water deficit conditions (Table 1). In comparison, the degree of dominance was higher than unity for the duration trait of cross (P1 × P2) under the water deficit condition and in cross (P3 × P4) under normal condition. In contrast, for 1,000-grain weight, the degree of dominance was less than unity in both crosses under both conditions.

Gene actions of generation means

The scaling test for adequacy of additive and dominance model, and genetic components of generation means showed the computed scales were significant for all studied characters, which indicate these traits are genetically control and depends on allelic interactions (Table 2). Findings also showed that the mean effect parameter (m) was highly significant for root, agronomic, and yield traits. Furthermore, additive gene action (d) played a major role in controlling the genetic variance of all traits. Additionally, dominance gene action (h) played a significant role in the inheritance of all traits in (P1 × P2) and (P3 × P4), except number of roots/plant and sterility % in (P1 × P2) under water deficit condition, panicle length in (P3 × P4) under normal condition, number of filled grains/panicle in (P3 × P4) under water deficit condition, number of panicles/plant in (P1 × P2) and (P3 × P4) under normal condition, plant height in (P1 × P2) under normal and in (P3 × P4) under water deficit conditions and duration in (P3 × P4) under normal, and in (P1 × P2) under water deficit and normal conditions (Table 2). Our findings revealed that the dominance gene action was greater than the additive gene action in the inheritance of among studied traits.

Table 2 Scaling test for adequacy of additive and dominance model and genetic components of generation mean of rice root, agronomic, and yield traits for the two studied crosses under both conditions.

Characters	Cross		Scaling test	Genetic components of generation mean	
			A	B	C	m	d	h	i	j	l	
Root length (cm)	I	N	1.06 ± 0.01**	0.72 ± 0.01	−11.75 ± 0.07**	27.33**	1.51**	22.92**	13.54**	0.16**	−15.32**	
D	−5.87 ± 0.01**	−5.27 ± 0.01**	−12.62 ± 0.02**	21.57**	0.78**	10.83**	1.47*	−0.29**	9.67**	
II	N	−4.58 ± 0.01**	0.56 ± 0.01	−22.24 ± 0.06**	27.58**	−4.47**	28.62**	18.21**	−2.57**	−14.19**	
D	−5.22 ± 0.02**	−2.45 ± 0.02**	−15.03 ± 0.09**	22.22**	−3.67**	12.26**	7.35**	−1.38**	0.33	
Root volume (cm3)	I	N	−28.71 ± 0.09**	−40.46 ± 0.02**	−69.35 ± 0.03**	70.56**	28.33**	60.52**	0.37	5.87**	68.80**	
D	−50.22 ± 0.02**	−24.77 ± 0.02**	−92.11 ± 0.09**	45.13**	2.27**	81.29**	17.76	−12.73**	57.23**	
II	N	−4.81 ± 0.06**	−10.86 ± 0.02**	−99.25 ± 0.05**	68.67**	−10.47**	136.88**	84.30**	3.02**	−68.62**	
D	−57.52 ± 0.04**	−35.71 ± 0.02**	−98.47 ± 0.08**	49.39**	−29.35**	70.03**	5.73	−10.90**	87.50**	
No. of roots/plant	I	N	−65.17 ± 0.09**	−12.16 ± 0.02**	−96.12 ± 0.06**	231.09**	9.91**	136.35**	18.95	−26.50**	58.37*	
D	−58.11 ± 0.05**	−126.11 ± 0.06**	−38.19 ± 0.07*	173.25**	36.41**	−6.75	−146.20**	33.91**	330.67**	
II	N	−39.07 ± 0.08**	−61.08 ± 0.09**	−147.14 ± 0.08**	244.02**	−23.27**	192.01**	47.15	11.00**	53.01	
D	23.42 ± 0.02**	39.32 ± 0.09**	134.93 ± 0.05**	143.41**	−12.42**	101.35*	52.63	−7.95**	1.61	
Root/shoot ratio (%)	I	N	−0.83 ± 0.001**	−0.56 ± 0.0001**	−1.20 ± 0.0011**	0.83**	0.26**	0.54**	−0.20**	−0.13**	1.60**	
D	−1.22 ± 0.003**	−5.87 ± 0.002**	−24.02 ± 0.010**	10.56**	2.97**	−14.35**	−23.82**	2.32**	30.42**	
II	N	−0.30 ± 0.001**	−0.29 ± 0.0001**	−0.25 ± 0.0013**	0.71**	−0.17**	−0.02**	−0.35**	−0.004**	0.96**	
D	−0.05 ± 0.001**	−0.04 ± 0.001**	0.03 ± 0.003**	2.70**	0.23**	0.17**	−0.15**	0.08**	0.28**	
Duration (day)	I	N	−13.21 ± 0.003**	−19.76 ± 0.01**	−46.18 ± 0.01**	123.99**	9.36**	31.88	13.10	3.27**	19.86	
D	−9.83 ± 0.03**	−3.25 ± 0.02**	−8.33 ± 0.05**	126.12**	1.95**	−5.89	−4.75	−3.29	17.83*	
II	N	−2.38 ± 0.08**	−6.31 ± 0.02**	−24.23 ± 0.05**	126.69**	6.01**	16.26**	15.41**	1.90**	−6.59**	
D	−10.20 ± 0.03**	−16.68 ± 0.02**	−24.65 ± 0.07**	132.13**	3.62**	1.12	−2.2	3.24**	29.12**	
Plant height (cm)	I	N	28.60 ± 0.05**	16.87 ± 0.08**	58.80 ± 0.09**	122.22**	3.13**	5.10	13.32**	5.86**	−32.15**	
D	−5.38 ± 0.03**	−10.71 ± 0.02**	1.68 ± 0.05	78.80**	3.70**	−14.28**	−17.78**	2.66**	33.88**	
II	N	3.62 ± 0.01**	12.25 ± 0.07**	22.87 ± 0.01**	124.75**	−3.62**	20.66**	−7.00	−4.31**	−8.87	
D	−1.45 ± 0.07	−1.33 ± 0.06	11.04 ± 0.03**	82.08**	3.25**	−20.52	−13.83	−0.05	16.62	
Panicle length (cm)	I	N	1.50 ± 0.01**	−1.63 ± 0.01**	−2.21 ± 0.07**	22.92**	0.63**	4.66**	2.07**	1.56**	−1.93**	
D	0.83 ± 0.011**	0.27 ± 0.020	0.48 ± 0.001**	20.80**	1.83**	3.74**	0.62	0.28**	−1.73*	
II	N	−1.07 ± 0.008**	−0.22 ± 0.010	0.20 ± 0.001	24.12**	−1.50**	2.25	−1.50	−0.42**	2.80	
D	−1.42 ± 0.004**	0.15 ± 0.006	−2.97 ± 0.034**	2,092**	−0.90**	2.61**	1.70**	−0.78**	−0.42	
No. of panicles/plant	I	N	−3.86 ± 0.01**	−1.80 ± 0.02**	−5.91 ± 0.04**	18.07**	2.12**	5.19**	0.25	−1.03**	5.41**	
D	−5.57 ± 0.06**	−4.98 ± 0.08**	−6.09 ± 0.05**	13.39**	3.05**	1.90	−4.46	0.29**	15.02**	
II	N	−3.30 ± 0.01**	−2.11 ± 0.01**	−6.41 ± 0.03**	19.15**	−2.15**	5.04**	1.00	−0.59**	4.41**	
D	−3.20 ± 0.05**	−1.97 ± 0.04**	−0.33 ± 0.03	16.09**	−3.46**	1.03	−4.84	−0.61**	10.01**	
No. of filled grains/panicle	I	N	23.31 ± 0.04**	−30.57 ± 0.08**	−59.55 ± 0.05**	100.68**	41.58**	23.27**	52.29**	26.94**	−45.02**	
D	−67.56 ± 0.04**	46.81 ± 0.09**	−132.12 ± 0.07**	62.57**	−35.60**	82.02**	111.55**	−57.18**	−90.80**	
II	N	−39.87 ± 0.07**	−7.46 ± 0.08**	−80.66 ± 0.07**	141.70**	−41.25**	68.44**	33.31**	−16.20**	14.02	
D	−27.90 ± 0.01**	−11.50 ± 0.09**	−47.25 ± 0.02**	108.74**	−16.36**	17.67	7.85	−8.20**	−31.54**	
1000-grain weight (g)	I	N	−3.00 ± 0.01**	−4.99 ± 0.01**	−3.33 ± 0.08**	25.65**	−3.62**	−3.12**	−4.65**	0.99**	12.65**	
D	−1.51 ± 0.01	−5.11 ± 0.02**	−8.14 ± 0.08**	21.78**	−1.58**	4.77**	1.50*	1.80**	5.13**	
II	N	−5.86 ± 0.01**	0.06 ± 0.01	−2.09 ± 0.08**	25.65**	1.18**	−4.41**	−4.37**	−2.63**	10.83**	
D	−6.09 ± 0.01**	−1.96 ± 0.01**	−2.84 ± 0.02**	21.90**	1.76**	−4.50**	−5.22**	−2.06**	13.28**	
Sterility %	I	N	−25.35 ± 0.03**	−3.10 ± 0.03**	28.05 ± 0.08**	28.60**	−13.25**	−23.70**	−56.50**	−11.12**	84.95**	
D	11.07 ± 0.06**	11.77 ± 0.07**	51.50 ± 0.07**	41.65**	−3.37**	9.20	−28.65**	−0.35**	5.80	
II	N	−0.08 ± 0.01	−17.13 ± 0.01**	−10.60 ± 0.08**	11.14**	7.42**	9.62**	−6.61**	8.25**	23.84**	
D	−2.47 ± 0.09**	0.41 ± 0.09	−32.90 ± 0.04**	17.70**	−2.71**	50.14**	30.84**	−1.44**	−28.78**	
Grain yield/plant (g)	I	N	−1.21 ± 0.02**	−9.83 ± 0.01**	−12.62 ± 0.08**	34.95**	7.53**	7.28*	1.57	4.31**	9.47**	
D	−4.32 ± 0.01**	−6.46 ± 0.01**	−11.23 ± 0.03**	24.32**	5.07**	5.30**	0.45	1.06**	10.33**	
II	N	−4.03 ± 0.01**	−30.21 ± 0.05**	−22.23 ± 0.09**	38.94**	9.70**	−7.23**	−11.75**	12.96**	45.75**	
D	−3.25 ± 0.01**	−5.81 ± 0.02**	−18.52 ± 0.05**	28.60**	−2.35**	16.51**	9.46**	1.28**	−0.40	
Note:

Cross P1 × P2, Giza178 × Sakha106; cross P3 × P4, Sakha104 × WAB56-104; N, normal and D, water deficit conditions. One asterisk (*) and two asterisks (**) indicates significant at 5% and 1% level of probability.

The additive × additive type of gene interaction (i) played a greater role in the inheritance of root length, root/shoot ratio, 1,000-grain weight and sterility % in the two crosses under normal and water deficit conditions. At the same time, it was positively significant for root volume (84.30), duration (15.41), number of filled grains/panicle (33.31) in (P3 × P4) under normal condition and grain yield/plant (9.46) under water deficit condition. Moreover, it was also positively significant for plant height (13.32), panicle length (1.70) and number of filled grains/panicle (52.29) in (P1 × P2) under normal condition, and for panicle length (2.07) and number of filled grains/panicle (111.55) under the water deficit condition in (P1 × P2). In contrast, it showed significant negative gene interaction for plant height (−17.78) and number of roots/plant (−146.20) in (P1 × P2) under water deficit condition and grain yield/plant (−11.75) in (P3 × P4) under normal condition (Table 2).

Additionally, additive × dominance type of gene interaction (j) has a significant role in the inheritance of root, agronomic, and yield traits, except duration trait in (P1 × P2) and plant height in (P3 × P4) under water deficit condition (Table 2). Meanwhile, dominance × dominance type of gene interaction (l) had an effective role in the inheritance of root volume and root/shoot ratio, number of roots/plant and 1,000-grain weight in both crosses under normal and stress conditions. While, it showed effective roles in the inheritance of root length, number of roots/plant, plant height, panicle length, number of filled grains/panicle and grain yield/plant in (P1 × P2), duration and sterility % in (P3 × P4) under normal and water deficit conditions. Similarly, root length and grain yield/plant in (P3 × P4) under normal condition and duration in (P1 × P2) under water deficit condition, number of filled grains/panicle in (P3 × P4) under water deficit condition, and sterility % in (P1 × P2) under normal condition (Table 2).

Genetic variance, genetic variability, heritability and genetic advance

Our study showed that the higher estimates of broad-sense heritability (h2b) were observed in the root, agronomic, and yield traits for (P1 × P2) and (P3 × P4) under normal and drought stress conditions, while low to moderate estimates of narrow-sense heritability (h2n) were detected under both conditions. The moderate estimates were noted for root length, root/shoot ratio, number of filled grains/panicle, and grain yield/plant in (P1 × P2) under normal condition; plant height, panicle length, and 1,000-grain weight in (P1 × P2) under water deficit condition, number of roots/plant, number of filled grains/panicle, and duration in (P3 × P4) under water deficit condition, duration in (P1 × P2) under both conditions (Table 3).

Table 3 Estimates of h2b, h2n, Gs and Gs % of root, agronomic and yield traits for the two studied crosses under both conditions.

Characters	Cross		Heritability %	Gs	Gs %	
		h2b	h2n	
Root length (cm)	I	N	80.23	41.95	11.71	42.84	
D	82.39	20.44	8.22	38.12	
II	N	89.01	22.39	8.04	29.17	
D	84.29	11.86	6.52	29.34	
Root volume (cm3)	I	N	94.60	20.90	27.37	38.82	
D	94.74	14.92	37.13	82.27	
II	N	95.95	9.67	11.04	16.08	
D	85.65	20.72	31.90	64.60	
No. of roots/plant	I	N	93.15	8.85	8.31	3.59	
D	90.17	29.42	91.49	52.81	
II	N	94.91	11.29	30.80	12.62	
D	78.34	30.60	65.98	37.83	
Root/shoot ratio (%)	I	N	66.53	35.18	0.81	98.32	
D	62.60	11.54	0.33	87.60	
II	N	60.46	27.22	0.67	94.49	
D	36.45	26.50	0.72	26.74	
Duration (day)	I	N	93.45	40.41	3.15	2.54	
D	90.60	44.40	55.83	44.72	
II	N	88.55	22.95	11.76	9.28	
D	88.55	37.44	45.96	34.78	
Plant height (cm)	I	N	92.77	23.69	18.74	15.33	
D	92.71	37.94	45.92	58.27	
II	N	91.96	29.48	29.81	23.89	
D	87.33	19.36	35.43	43.16	
Panicle length (cm)	I	N	80.47	27.40	8.34	36.41	
D	88.63	44.95	15.78	75.90	
II	N	87.27	26.83	17.76	73.63	
D	86.38	4.86	1.67	7.98	
No. of panicles/plant	I	N	85.51	11.36	5.95	32.92	
D	86.25	14.34	11.68	87.24	
II	N	89.67	11.12	6.78	35.41	
D	85.41	16.58	13.22	82.18	
No. of filled grains/panicle	I	N	74.57	39.66	34.25	34.02	
D	84.29	24.67	61.75	98.68	
II	N	94.23	11.34	15.38	10.86	
D	80.41	34.57	59.54	54.75	
1,000-grain weight (g)	I	N	90.19	20.88	7.67	29.91	
D	87.23	42.25	16.11	74.00	
II	N	90.07	20.69	7.83	30.55	
D	90.33	19.06	7.81	35.67	
Sterility %	I	N	89.51	5.44	4.63	16.21	
D	90.60	33.54	38.98	93.58	
II	N	86.74	16.74	7.71	69.22	
D	83.75	21.29	17.29	97.64	
Grain yield/plant (g)	I	N	97.87	6.42	5.86	16.78	
D	89.88	20.71	8.35	34.36	
II	N	81.23	6.9	2.61	6.71	
D	93.04	17.58	9.43	32.97	
Note:

Gs, genetic advance and G.S %, expected genetic advance. Cross P1 × P2, Giza178 × Sakha106; cross P3 × P4, Sakha104 × WAB56-104; N, normal and D, water deficit conditions.

The knowledge of genetic advances (Gs) produced by applying selection pressure to a population is helpful in developing an efficient breeding program. High estimates of Gs and Gs % expressed as percentages of mean were found in the root, agronomic, and yield traits of two crosses under normal and water deficit conditions, except the number of roots/plant, and duration in (P1 × P2), duration and grain yield/plant in (P3 × P4) under normal condition, and panicle length in (P3 × P4) under water deficit condition, which exhibited low values (Table 3). Additionally, most of the studied traits showed higher estimates of h2b thereby higher genetic advance.

Discussion

This study implies mechanistic insights the variable level of genetic dominance that controls important agronomic traits in rice populations under water deficit condition. This study further explored gene actions, heritability, genetic advances and heterosis of several physiological and agronomic traits in rice polutions under water deficit conditions. The findings of this study disclose a new platform of future phenotypic and genotypic association studies in rice. These insights could be useful to the rice breeders and farmers for improving water deficit stress tolerant in rice cultivars.

Mean performance and heterosis

Our findings indicated that the means of six populations in two studied crosses (first parent (P1), second parent (P2), first-generation (F1), second-generation (F2), first backcross (BC1) and second backcross (BC2)) showed a wide range among two parents in most of the studied traits under normal and stressful conditions. Moreover, F1s, F2s, BC1 and BC2 under both conditions were higher than the highest parent for root, agronomic, and yield traits in the two crosses, which suggests that overdominance could play a major role in the inheritance of these traits. Furthermore, the desirable transgressive segregation in a successive generation provided the desirable complementary genes and epistatic effects, which were coupled in the same direction to maximize such traits under consideration. However, the F1 mean values for duration, number of filled grains/panicle and 1,000-grain weight were sometimes intermediate between the two parents, suggest that these traits are might be partially controlled by the cross itself.

Significant positive estimates of heterosis and heterobeltiosis among physiological and agronomic traits and its components in two studied crosses, which revealed the over-dominance plays an important role in inheritance of those traits. These findings were supported by several studies (Ahmadikhah, 2018; Hossain et al., 2021; Kamara et al., 2017; Tiwari et al., 2019), wherein it has also been suggested that these aforementioned crosses could be used to explore the heterotic effects in hybrid rice breeding programs with water deficit conditions (Concepcion et al., 2015; Guimarães, Stone & Silva, 2016; Hassan, 2017; Hastini et al., 2021; Potkile et al., 2018). Furthermore, significant heterosis percentages were recorded for these traits in studied crosses, which was corroborated by previous studies (Gaballah et al., 2021; Ouyang, Li & Zhang, 2022; Sakran et al., 2022).

Degree of dominance and genetic components of generation mean

Degree of dominance, additive and dominance gene action

The degree of dominance was higher than unity (±1.0) for all physiological and agronomic traits except 1,000-grain weight, which was less than unity in both crosses under both conditions. These results indicate that complete dominance may have significant role in the inheritance of such traits. The ratios between zero and ±1.0 suggest that partial dominance effects in inheritance of such characters. These results were reported by the previous studies (Guilengue et al., 2020; Nihad et al., 2021).

In our study, additive gene action (d) showed a significant role in inheritance of physiological and agronomical traits. Moreover, (h) played a greater role in the inheritance of all traits in the two crosses, except for number of roots/plant and sterility % in (P1 × P2) under water deficit conditions, and it was greater than (d) in the inheritance of among studied traits. The additive gene actions were detected by several studies in plants (Ahmadikhah, 2018; Ganapati et al., 2020; Guimarães, Stone & Silva, 2016; Kamara et al., 2017). However, Kamara et al. (2017), found a link of additive effects in regulating the expression of plant height, number of panicles/plant, number of filled grains/panicle, sterility % and grain yield/plant, while non-additive gene actions were important for the inheritance of 1,000-grain weight. Additionally, Lukman & Suyamto (2017) found that additive and non-additive gene effects significantly control the duration and plant height (Lukman & Suyamto, 2017). In our present study, such type of gene actions was confirmed in rice.

Non-allelic interaction

The additive × additive types of gene interactions (i) were played a major role in the inheritance of root length, root/shoot ratio, 1,000-grain weight, and sterility % traits in the two crosses under both conditions. These findings imply that the examined traits were inherited in these crosses mostly due to additive gene effects. Additive gene effects can be exploited in early generations because the dominance effects are also non-significant and lower in magnitude than these additive effects. These findings agreed with those of Abebe, Alamerew & Tulu (2019), Hastini et al. (2021) and Lukman & Suyamto (2017). In addition, additive × dominance type of gene interactions (j) were played a significant role in the inheritance of all the traits that were investigated, except the duration trait in (P1 × P2) and plant height in (P3 × P4) under water deficit conditions. Also, (l) played an effective role in the inheritance of root volume and root/shoot ratio, number of panicles/plant and 1,000-grain weight traits in both crosses under both conditions. Moreover, it played key roles in controlling the genetic variance among root, agronomic, and yield traits for two crosses under both conditions; similar results were reported by Concepcion et al. (2015) and Potkile et al. (2018).

In contrast, dominance gene action, additive × dominance and dominance × dominance types of gene interaction showed highly significant values, indicating that these factors are significant contributors to the variation of generation means, which were involved in the inheritance of such characters, similar results were reported by Hassan (2017) and Santosh et al. (2014). Since additive gene effects were non-significant for these characters, a simple selection procedure in the early generations may not contribute significantly to improving these characters. The additive components in these traits can be successfully exploited through the pedigree method of selection because of the major contribution of additive gene effects in late generations of segregating populations. However, our study corroborates to the other reports (Kumar et al., 2014; Ouyang, Li & Zhang, 2022). We found different gene interactions under water deficit conditions that might be the linkage between the genes displaying digenic epistasis effects and the relative frequencies of different combinations of genotypes in the progeny of a cross or self. Hints the epistasis depended on these gene combinations, the nature, and extent of the epistasis change under normal and water deficit conditions. Therefore, the magnitudes and directions of the various parameters can only be interpreted in relative terms, and we can say little about their maximum values because they may be affected by allele dispersion or the ambidirectionality of the intra-and inter-genic effects. Allele dispersion is particularly awkward for interpreting the type of epistasis because it can affect both the magnitudes and the signs of additive × additive and additive × dominance. Further, it is generally not possible to determine whether the effects of the various genes are equal or not. Consequently, it is rarely possible to distinguish complementary from recessive epistasis or duplicate from dominant epistasis. Thus, with polygenic traits, epistasis can only be classified as either predominantly duplicate or predominantly complementary, the distinction being based solely on the relative signs of the dominance and the dominance × dominance (Kearsey & Pooni, 1996).

Genetic variance, genetic variability, heritability and genetic advance

The dominance genetic variance (1/4 H) was higher than the additive genetic variance (1/2 D) for all studied traits, demonstrating that the expression of all the examined traits was dominated by a non-additive component of genetic variance in both crosses under both conditions. These findings were closely related to the previous studies (Concepcion et al., 2015; Hassan et al., 2018). Low to moderate estimates of narrow sense heritability (h2n) were detected under both conditions for among studied traits. The moderate estimates were recorded for some traits under water deficit conditions, sometimes for (P1 × P2) and others for (P3 × P4) crosses. This juncture leads to the conclusion that these traits can be improved through conventional breeding methods and selection can be effective mostly in later generations, similar results were found by Abebe, Alamerew & Tulu (2019) and Farooq et al. (2019). High estimates of GS % expressed as mean percentages were found in all the studied traits in both crosses under both conditions, which were corroborated by the previous study (Sakran et al., 2022). In addition, most of the studied traits had high estimates for h2b and therefore high GS, suggested that those traits are either simply inherited or governed by a few major genes, or that additive gene effects played important roles in the inheritance of these traits (Ouyang, Li & Zhang, 2022). On the other hand, low heritability was reported for plant height and yield of rice plants (Karavolias et al., 2020; Tiwari et al., 2019). On the contrary, high broad-sense heritability (h2b) was reported for plant height, 100-grain weight (Farooq et al., 2019), and number of panicles/plant (Hastini et al., 2021). Additionally, Hassan et al. (2018) also found high h2b and high GS for different traits, such as duration, plant height, number of filled grains/panicle, 1,000-grain weight, sterility % and grain yield/plant.

Conclusion

This study unveils the genetic background of each genotype that showed significant differences both in water deficit and normal conditions. Differences in physiological and agronomical traits among the populations indicate that variations in genetic mechanisms contributed to the rice generations in response to water deficit stress. This study was further supported by the Sakha104 × WAB56-104 (P3 × P4) had the best values under water deficit conditions, which showed the new genetic structure produced by directional hybridization and how it could solve the problem of water deficit stress. Additionally, the heterosis produced during hybridization in excellent genotypes that overcome the defects of some traits caused by water deficit stress. Furthermore, additive and dominance gene action played an effective role in the inheritance of all the studied traits in the two crosses, while the results revealed that the dominance gene action was greater than additive gene action in the inheritance among studied traits under both conditions. This study further indicates that the non-allelic type of gene interactions (i, j and l) have effective roles in the inheritance of most traits in two crosses under both conditions. Low narrow-sense heritability with high genetic advance indicates the control of non-additive genes, so that hybridization and then selection may be effective for those traits. Therefore, these findings might be useful to the rice breeders and farmers for future phenotypic and genotypic association studies, and will encourage to develop drought stress tolerance rice cultivars.

Supplemental Information

Supplemental Information 1 Supplementary Tables.

Click here for additional data file.

We want to express our deep and sincere gratitude to workers at the farm of Rice Research Agricultural Station, Sakha, Kafr El-Sheikh, Egypt.

Abbreviations

D Additive

H Dominance

E Environmental

GS Genetic advance

GS % Expected Genetic advance

h2b Broad-sense heritability

h2n Narrow-sense heritability

m Mid-parent value

d Additive gene action

h Dominance gene action

i Additive × additive type of gene interaction

j Additive × dominance type of gene interaction

l Dominance × dominance type of gene interaction

1/2 D Additive genetic variance

1/4 H Dominance genetic variance

MP Heterosis as a deviation from mid-parents

BP Heterosis as a deviation from better-parents (heterobeltiosis)

Additional Information and Declarations

Competing Interests

Author Contributions

Data Availability

The authors have no conflict of interest to disclose.

Hamada M. Hassan conceived and designed the experiments, performed the experiments, analyzed the data, prepared figures and/or tables, authored or reviewed drafts of the article, and approved the final draft.

Adel A. Hadifa conceived and designed the experiments, performed the experiments, analyzed the data, prepared figures and/or tables, authored or reviewed drafts of the article, and approved the final draft.

Sara A. El-leithy conceived and designed the experiments, analyzed the data, prepared figures and/or tables, authored or reviewed drafts of the article, and approved the final draft.

Maria Batool conceived and designed the experiments, analyzed the data, prepared figures and/or tables, authored or reviewed drafts of the article, and approved the final draft.

Ahmed Sherif conceived and designed the experiments, analyzed the data, prepared figures and/or tables, authored or reviewed drafts of the article, and approved the final draft.

Ibrahim Al-Ashkar conceived and designed the experiments, prepared figures and/or tables, authored or reviewed drafts of the article, and approved the final draft.

Akihiro Ueda conceived and designed the experiments, authored or reviewed drafts of the article, and approved the final draft.

Md Atikur Rahman performed the experiments, authored or reviewed drafts of the article, and approved the final draft.

Mohammad Anwar Hossain conceived and designed the experiments, authored or reviewed drafts of the article, and approved the final draft.

Ayman Elsabagh conceived and designed the experiments, authored or reviewed drafts of the article, and approved the final draft.

The following information was supplied regarding data availability:

The raw data is available in the Supplemental File.

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
