# Peer review of "Variable level of genetic dominance controls important agronomic traits in rice populations under water deficit condition"

_PeerJ, doi:10.7717/peerj.14833_

## Round 0.1 · original submission · Major Revisions

Dear Authors
According to the reviewer's comments, this manuscript cannot be accepted for publication. It needs a major revision to be reconsidered for publication. The authors are invited to revise the paper considering all the suggestions made by the reviewers. Please note that requested changes are required for publication.

Additional comments:

1- The manuscript is very weak. The title of the manuscript must be shortened.

2- Several parts of the paper are not scientifically sound and need to be rewritten and developed, including the introduction, experimental design and material & method, results, and discussion sections.

3- The presented figures are a very bad design and resolution, it needs more creativity and improved resolution.

4- The author's affiliation differs on the first page and the manuscript's first page.

5- Authors have to attach along with the revised version of photos for the experiments.

6- In addition to the comments relating to the science being reported, there are significant concerns about the manuscript's grammar, usage, and overall readability. Therefore, revise the text to fix grammatical errors and improve the text's overall readability. We suggest you have a fluent English-language speaker thoroughly copyedit your manuscript for language usage, spelling, and grammar. If you do not know anyone who can do this, PeerJ can provide language editing services.

Reviewer 1 ·

Basic reporting

I have checked the manuscript. It is well written but need minor revision. It is better to bring the table of variance analysis and also The quality of the figures is not good and should be improved.
Regards

Experimental design

It is well done

Validity of the findings

It is well done

Reviewer 2 ·

Basic reporting

The manuscript is based on a good main idea and is at a level to contribute to the literature scientifically. However, it is understood that it is difficult to explain the desired goal. It should almost be rewritten. The script is very weak. The narration bores the reader. In particular, the results and discussion must change throughout. They should be combined if necessary and appropriate for your journal.
The English language is difficult to understand and contains grammatical errors, and a professional review is recommended.
The writing rules of the journal were not observed especially in the presentation of references. It should be corrected.
Figures should be reorganized to be more understandable. Authors made two hybrids, but it is not clear which one was shown where.
Although the abbreviations made in the text and those in the figure or table are not compatible, they should be presented in a single order and added to the abbreviations section. It would be better if different encodings were made for each hybrid.

Experimental design

The experimental design and material method section need to be developed. Detailed explanations about this section have been added to the pop-up notes on the pdf file.

Validity of the findings

The expression is quite complex. Must be rewritten. If the editor approve, it should be combined with the discussion. Discussing the results as "consistent/not compatible with previous studies" reduces the scientific value of the manuscript. Why did you find different gene interactions under drought stress? Discuss this mechanism.

Annotated reviews are not available for download in order to protect the identity of reviewers who chose to remain anonymous.

Reviewer 3 ·

Basic reporting

The manuscript provides a fairly robust dataset on exploring the genetic differences, genetic advances, heritability, and heterosis in rice populations under water deficit conditions. The authors put effort into crossing different genotypes that varied in their drought tolerance and developed F1, F2, and BC1 and BC2 populations. I suggest accepting the manuscript following major revision.
The manuscript needs major English editing, the abstract starts with a grammatical error “This investigation was aimed to ……”. Long sentences as in lines 32-41 which is one sentence in 10 lines. All manuscript needs major English editing.
The abstract lacks Ms&Ms details as the applied irrigation regimes…
The symbol (×) should be used instead of the letter (x) in different interactions as “additive x additive, additive x dominance, Giza178xSakha106 …..”
All mentioned results in the abstract were on gene action. This should be resumed and the other aspects as mean performance of different populations under different irrigation regimes, the impact of drought stress, genetic advance, heritability, and heterosis should be presented.
The introduction needs to be improved, the hypothesis needs to be clarified. The negative impact of drought stress on rice's physiological and agronomic traits, the importance of genetic parameters need to be expanded and improved.
Ms&Ms
Lines 115-118: The measured traits should not be presented under the subtitle “Crossing and development of F1 and F2 populations”. But should be presented under a separate subtitle with more details on their measurement, particularly root traits.
Line 120-122 how the author identified these genotypes based on their tolerance (tolerant or sensitive) to water deficit conditions. The mentioned Table 1 in line 122 is not appropriate, it does not present the difference among selected parents. But it presents the mid-parent and better-parent heterosis for the evaluated root and agronomic traits. I did not find the supplementary data as presented in line 122 (S1)
Line 130-133 More details should be added on the agricultural practices, N, P, K fertilization rates, water irrigation amount….. Also, more details are needed on the number of rows in each plot, row length, and distance among plants, ….

The results are inadequately presented and should be improved. The mean performance of studied root and agronomic traits of the evaluated populations derived from two crosses is poorly presented and needs major improvement. Also, the other subtitles of the result section need to be improved. The four parents Giza178, Sakha106, Sakha104, and WAB56-104 were used for developing F1, F2, BC1, and BC2. The complete name of parents could be presented in the Ms&Ms alongside codes and then throughout the manuscript, they could be used as P1×P2 instead Giza178xSakha106 and P3×P4 Sakha104×WAB56-104.
The figures need more effort to improve their presentation. Both crosses could be presented in the same figure for the same trait.
The caption of Fig. 2 needs more clarification. The presented codes in the figures should be explained in the titles not mention “ P1: female parent (Giza178 and Sakha104)” but P1C1:….. P2C1:…..
Fig. 2 is cited in Line 197 after citing Fig.3-5 in line 194

The references should be revised, some journals are abbreviated as s. Intl J Farm & Alli Sci (line 464) While most journals are not abbreviated.

Experimental design

Adequate

Validity of the findings

Adequate

·

Basic reporting

It is stated how the research fills an identified knowledge gap. Water scarcity is one of the major environmental factors hindering rice production, so the research question lowering the gap causing by water deficit to avoid climatic changes.
Gives good background for his work , except root trails
Cited recent literature according the importance of root parameters and trails.
-Needs to re-arrange.
-Omit the ancient citation down 2015.

Experimental design

Line 188; statistical program had no cleared citation, in this case SPSS or SAS is better for more significant differentiation.
ANOVA Tables with all the introducing traits for F1, F2 … with parents used should be tabulated.

Validity of the findings

Meaningful of findings had hesitating results needs more analysis tools in combining ability

Additional comments

I commend the authors for their extensive data set, compiled over many years of detailed fieldwork, which gave an evaluation for gene action, heritability, genetic advance and heterosis of rice root, agronomic, and yield component traits under water deficit conditions as a global needs for saving water resources and for rice breeders under drought stress conditions.
But Gaballah et al., 2022 (Plants, 11, 66) showed the estimation values of better parent heterosis (BPH) were varied among the cross combinations, why you use your tools far from these recent traits.

---

## Round 0.2 · Minor Revisions

Dear Author

Thanks for your efforts to improve the manuscript. However, there are still significant concerns about the manuscript's grammar, usage, and overall readability. Therefore, we suggest you have a fluent English-language speaker thoroughly copyedit your manuscript for language usage, spelling, and grammar.

Also, The resolution of Figure 2 still needs to be improved.

Reviewer 1 ·

Basic reporting

Dear Editor
All the requests of the reviewers have been done and therefore it is acceptable.
Regards

Experimental design

good

Validity of the findings

goof

Additional comments

Dear Editor
All the requests of the reviewers have been done and therefore it is acceptable.
Regards

Reviewer 2 ·

Basic reporting

Dear Editor,
The authors have made efforts to improve the manuscripts taking into account the comments of all reviewers. However, I think that I should ask for very minor corrections again. Because the research idea is very good and it should be explained as the best version. I'm listing the minor changes I want below.

1. Please choose one throughout the text: "water deficit" or "drought". If you choose "water deficit" correct it in the title as well.
2. Line 151: "however" changed with "therefore"
3. Line 285: isn't it complicated to say P1 in P3 x P4? Just P3 is enough, it's already parent.
4. In Tables, show hybrids with their own codes instead of "I" and "II". All text maintains the same flow and integrity is ensured.
5. This is a somewhat general comment. But you wrote about the drought. I wonder if it can be attributed to changes in gene expressions under environmental stresses? You can add a more focused comment with current literature. Although these studies are being done at the molecular level, it may offer you the oppor-tunity to interpret genes better?

The manuscript can be accepted after the changes I have indicated. I don't need to see it again. I wish the authors success in their new research.

Experimental design

none

Validity of the findings

none

Reviewer 3 ·

Basic reporting

The manuscript still has spelling and grammatical errors and needs major English editing (under water deficit conditions for among studied traits, proteinper, had highly significantly positive …..).
The resolution of Figure 2 is still bad and needs to be improved

Experimental design

Adequate

Validity of the findings

Adequate

·

Basic reporting

The authors did my suggestion in right way

Experimental design

Done in clear way but not sufficient

Validity of the findings

Done

Additional comments

Citations need more updating

---

## Round 0.3 · accepted · Accept

Dear Authors
I am pleased to inform you that after the last round of revision, the manuscript has been improved a lot, and it can be accepted for publication.

Congratulations on the acceptance of your manuscript, and thank you for your
interest in submitting your work to PeerJ.